# Structure of the p53/RNA polymerase II assembly

Shu-Hao Liou[1], Sameer K. Singh[1], Robert H. Singer[1,2], Robert A. Coleman[1]✉ & Wei-Li Liu[1,3]✉

The tumor suppressor p53 protein activates expression of a vast gene network in response to stress stimuli for cellular integrity. The molecular mechanism underlying how p53 targets RNA polymerase II (Pol II) to regulate transcription remains unclear. To elucidate the p53/Pol II interaction, we have determined a 4.6 Å resolution structure of the human p53/Pol II assembly via single particle cryo-electron microscopy. Our structure reveals that p53's DNA binding domain targets the upstream DNA binding site within Pol II. This association introduces conformational changes of the Pol II clamp into a further-closed state. A cavity was identified between p53 and Pol II that could possibly host DNA. The transactivation domain of p53 binds the surface of Pol II's jaw that contacts downstream DNA. These findings suggest that p53's functional domains directly regulate DNA binding activity of Pol II to mediate transcription, thereby providing insights into p53-regulated gene expression.

[1] Gruss-Lipper Biophotonics Center, Department of Anatomy and Structural Biology, Albert Einstein College of Medicine, Bronx, NY, USA. [2] Janelia Research Campus, Howard Hughes Medical Institute, Ashburn, VA, USA. [3]Deceased: Wei-Li Liu. ✉email: robert.coleman2@einsteinmed.org; lily.w.liu@gmail.com

Precise transcription of protein-encoding genes requires RNA polymerase II (Pol II) along with Mediator and six general transcription factors (GTFs, including TFIIA, TFIIB, TFIID, TFIIE, TFIIF, and TFIIH) to form the pre-initiation complex (PIC)[1–4]. The PIC occupies the transcription start site where it melts promoter DNA, leading to Pol II transcription initiation[5,6]. PIC formation is inherently inefficient in the absence of transcriptional activators, due to the limited accessibility of GTFs to target gene promoters. To rapidly respond to diverse stimuli[7,8], it is well established that transcriptional activators bind their consensus sequences on target gene promoters to aid PIC assembly and stimulate transcription[9,10]. The human p53 protein is a potent transcriptional activator that turns on diverse gene expression programs to regulate cellular processes (e.g., cell cycle arrest, DNA repair and apoptosis) for tumor inhibition[11,12]. p53 is promptly activated upon various stress signals that include DNA damage, oncogene activation and hypoxia[12]. Consequently, p53 plays a critical role in cellular integrity to prevent transformation into a cancerous state. Therefore, it is important to gain deeper insights into the molecular mechanism underlying how p53 regulates gene expression via emerging advanced approaches.

A battery of evidence demonstrates that p53 exhibits multiple activities to activate transcription[13]. Biochemical studies have shown that p53 binds consensus sequences (i.e., response elements, REs) on target gene promoters to directly turn on transcription[14]. p53 can also directly bind and recruit several components of the PIC (e.g., Mediator, TFIIB, TFIID, TFIIH) to synergistically promote the assembly on the promoter[15–21]. In addition to transcription initiation, our recent work demonstrates that the interaction of p53 and Pol II directly enhances the elongation efficiency of Pol II[22]. These activities could be attributed to p53's ability to structurally regulate target co-factors. Our previous reports utilizing cryo-EM and single molecule fluorescence microscopy have documented that p53 induces conformational changes of TFIID to aid recruitment to target gene promoters[17,23]. p53 can also introduce structural changes of Mediator to transition Pol II from the poised state into the elongation phase[18]. Our recent structural work reveal that the p53/Pol II association permits the Pol II DNA-binding clamp to adopt a closed state[22]. Collectively, these studies suggest that the interaction between p53 and its target factors introduce specific structural features that could help direct transcription activation. Hence, it is essential to unravel the detailed structural basis on how p53 interacts with these key transcription factors to regulate transcription.

The p53 protein contains the N-terminal transactivation/proline-rich domains (i.e., TADs and PRD), the DNA-binding core domain and the C-terminal oligomerization domain (i.e., OD) along with the regulatory domain (i.e., CTD)[24] (Fig. 1a). The intrinsically disordered transactivation domains (i.e., TAD1 and TAD2) can be structurally stabilized to form a helical structure upon binding to p53 interactors (e.g., MDM2[25], replication protein A[26] and CBP/p300[27]). The central core domain is structurally composed of a β-sheet sandwich scaffold and a helix that permit sequence-specific DNA binding to p53 REs. The C-terminal oligomerization domain allows p53 to form dimers and tetramers that facilitate binding to its REs on target genes via the core domain[28–30]. Thus far, due to several inherent structurally disordered domains (e.g., TADs, proline-rich domain and regulatory domain), the high-resolution tertiary structure of full-length p53 remains enigmatic[31–35]. Overall, these functional domains orchestrate p53's binding to target genes and its regulators in order to direct p53-regulated transcription and maintain genomic stability. Therefore, it would be intriguing to uncover how these domains of p53 structurally engage its key target transcription factors, such as Pol II[36]. To this end, we have

previously determined an 11 Å resolution cryo-EM structure of p53/Pol II[22], revealing that the p53 core domain selectively occupies regions within Pol II where the upstream DNA and the elongation factor DSIF bind. However, due to limited resolution[22], it has been difficult to pinpoint the precise positioning of p53's functional domains that interact with Pol II. Thus, armed with advances in cryo-electron microscopy and software, we set out to tackle this challenging task, seeking to capture the p53/Pol II structure at unprecedented resolution and gain detailed structural insights.

Herein, we present a 4.6 Å resolution cryo-EM three-dimensional (3D) structure of the human p53/Pol II co-complex. The structure reveals that the core domain of p53 is located at the top of the Rpb2 protrusion and the upstream DNA binding site of Pol II. This interaction introduces a large structural change of the Pol II clamp coiled-coil into a further-closed conformational state. This closure of the clamp might negatively regulate Pol II's interaction with DNA. In addition, a cavity was identified between the interface of p53's key DNA-binding helix and the Rpb1 clamp, posing a possibility for p53 to engage response elements within target genes. Furthermore, the C-terminal oligomerization domain of p53 is exposed on the top of the core domain and not in contact with Pol II, indicating a potential to form oligomeric p53 and/or gain post-translational modifications[13]. It is of note that the N-terminal transactivation domains of p53 bind the Pol II jaw that can contact downstream DNA and TFIIH during PIC formation[4,6]. Collectively, these findings indicate the unique positioning of p53's functional domains allow p53 to directly regulate the DNA binding activity of Pol II. The select locations of p53 within Pol II may also imply the de novo recruitment of additional monomeric p53 and/or other transcriptional factors (e.g., GTFs and co-factors) to promote transcription activation, thereby providing insights on p53-regulated Pol II transcription.

## Results

**Structural architecture of the p53/Pol II assembly.** To structurally dissect the detailed interaction between human Pol II and p53, we immunopurified Pol II from HeLa nuclear extracts using a monoclonal antibody against the C-terminal domain of the Rpb1 subunit of Pol II. The Pol II bound affinity resin was subsequently washed and incubated with an excess of recombinant p53 and a promoter fragment. Unbound p53 and promoter fragment were removed from the affinity resin via extensive washing. The p53/Pol II co-complex was eluted from the affinity resin as described in our previous studies in the absence of crosslinking[22]. Armed with current advanced single particle cryo-electron microscopy, we collected 776,710 single-particle images and have determined the 3D reconstruction of p53/Pol II at 4.6 Å global resolution (Fig. 1b, Table 1, Supplementary Figs. S1 and S2, Supplementary Movie 1 and Supplementary Movie 2). We used masked classifications to generate the maps due to the highly flexible nature of our p53/Pol II co-complex (Supplementary Fig. S1). Consistent with our previous ~11 Å resolution p53/Pol II structure[22], the large prominent extra cryo-EM density represents the core domain/C-terminus of p53 (Fig. 1b and Supplementary Fig. S3B). A relatively small but prominent extra density was identified in between the Pol II jaw and assigned as the N-terminus of p53 (Fig. 1b, left panel and Supplementary Fig. S4A), which was not captured in our previous work[22].

To further gain structural insights, we have determined the molecular structure of p53/Pol II based upon our 3D reconstruction and the atomic structure of the human closed Pol II (PDB: RCSB 5IYA[6]) via Phenix[37] (Fig. 1b, c). To better fit our p53/Pol II structure, the Rpb1 subunit of Pol II (spanning 158 to 179 aa) was

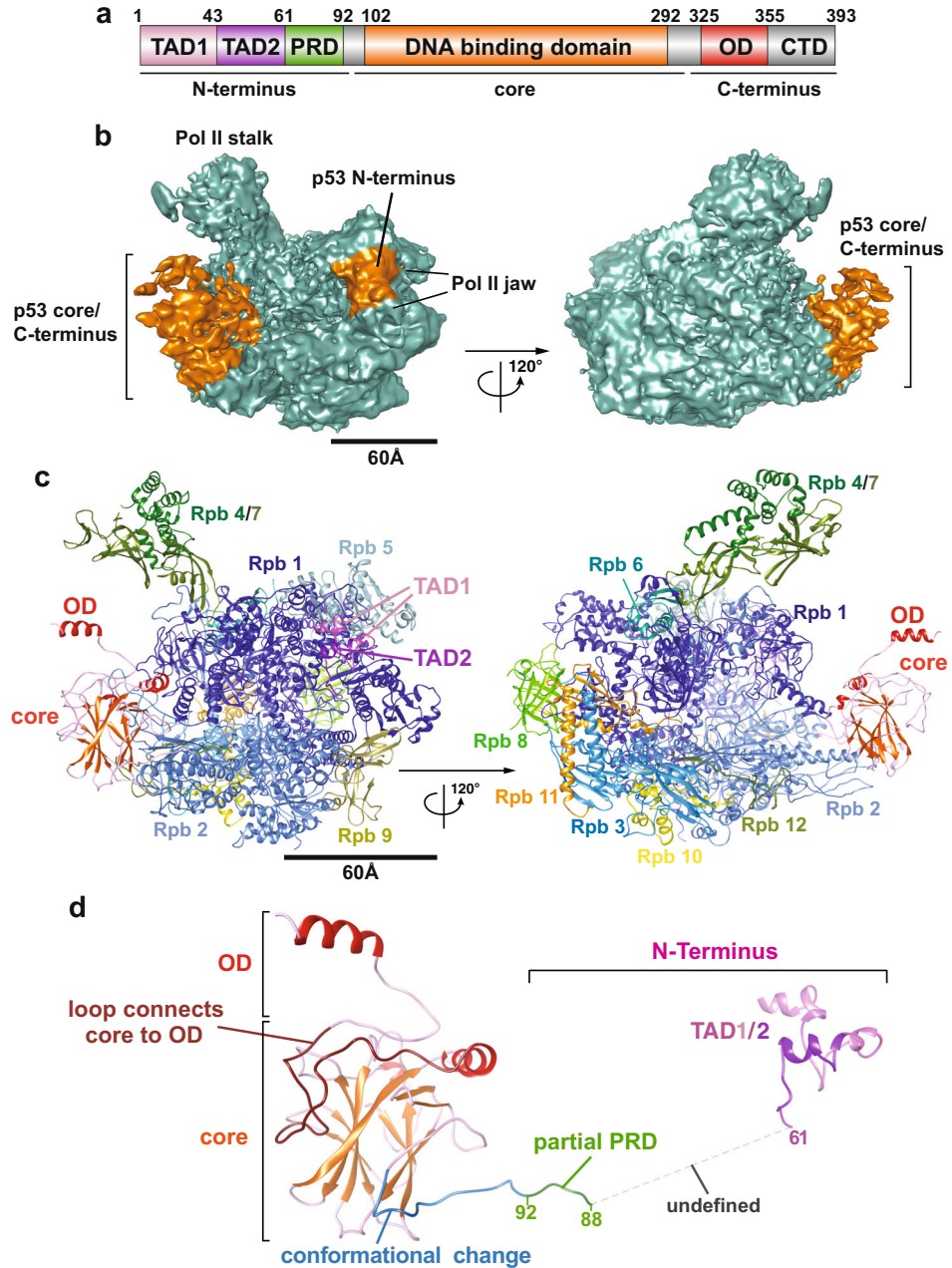

**Fig. 1 3D reconstruction and molecular structure of the p53/Pol II co-complex. a** Schematic representation of the domain structure of p53. Delineation of the transcription activation domains (TADs, pink/lilac), proline-rich domain (PRD, green), DNA binding domain (core domain, orange), oligomerization domain (OD, red), and C-terminal regulatory domain (CTD, gray) are presented. **b** 3D reconstruction of p53/Pol II is shown in two different views. The cryo-EM density representing the p53 protein is highlighted in orange. The Pol II density is colored teal with two canonical structural features indicated. The scale bar represents 60 Å. **c** Two different views of the p53/Pol II molecular structure are shown. Key functional domains of p53 and Pol II's subunits are indicated and color-coded. **d** Overall molecular structure of the p53 protein with main domain features as indicated is shown. A portion of proline-rich domain (PRD, 62-88 aa) is unresolved (due to insufficient cryo-EM densities), which is presented as a gray dash line that connects TADs and PRD/core domain. The N-terminal region of the p53 core domain (94-109 aa, blue loop) undergoes a conformational change compared to the previous core domain crystal structure (PDB 3TS8)[34].

further de novo constructed by Rosetta[38]. Atomic structures of monomeric p53's DNA-binding core and oligomerization domains were fit into the prominent p53 cryo-EM density (Fig. 1b, c and Supplementary Fig. S3B), by utilizing the crystal structure of an engineered human p53 protein (PDB: RCSB 3TS8)[34]. The p53 core domain specifically interacts with the region of Pol II's Rpb1 and Rpb2 subunits that engage upstream DNA (i.e., DNA binding cleft of Pol II)[6]. In addition, the

structure of the C-terminal oligomerization domain is exposed on top of the core domain and distant from the core domain/Pol II contact surfaces (Fig. 1c). This unique position poses a possibility for subsequent oligomerization of monomeric p53 when bound to Pol II. Importantly, we were able to define the molecular model of p53's N-terminal transactivation domains (TADs) within the smaller p53 cryo-EM density (Fig. 1b and Supplementary Fig. S4A) via fitting with the NMR structure (PDB: RCSB

**Table 1 Cryo-EM data collection, refinement and validations statistics.**

| | p53-Pol II (EMDB-22294) (PDB 6XRE) |
|---|---|
| **Data collection and processing** | |
| Magnification | 38168 |
| Voltage (kV) | 300 |
| Electron exposure (e–/Å²) | 1.16 |
| Defocus range (μm) | −0.5-−2.5 |
| Pixel size (Å) | 0.655 |
| Symmetry imposed | N/A |
| Initial particle images (no.) | 776710 |
| Final particle images (no.) | 92522 |
| Map resolution (Å) | 4.6 |
| FSC threshold | 0.143 |
| Map resolution range (Å) | 3–11 |
| **Refinement** | |
| Initial model used (PDB code) | 5IYA/3TS8 |
| Model resolution (Å) | 4.6 |
| FSC threshold | 0.143 |
| Model resolution range (Å) | 3–11 |
| Map sharpening $B$ factor (Å²) | 6.216 |
| Model composition | |
| Non-hydrogen atoms | 34,262 |
| Protein residues | 4289 |
| Ligands | 9 |
| $B$ factors (Å²) | |
| Protein | 12.66 |
| Ligand | 16.00 |
| R.m.s. deviations | |
| Bond lengths (Å) | 0.011 |
| Bond angles (°) | 1.309 |
| Validation | |
| MolProbity score | 1.75 |
| Clashscore | 4.76 |
| Poor rotamers (%) | 0.24 |
| Ramachandran plot | |
| Favored (%) | 91.53 |
| Allowed (%) | 7.89 |
| Disallowed (%) | 0.59 |
| CC(box)/ CC(mask) | 0.62/0.48 |

2L14)[27] and a partial de novo construction (See Methods). TADs are anchored in between the Pol II jaw (composed of Rpb1 and Rpb5) where downstream DNA traverses[6] (Fig. 1c, left panel).

The global molecular structure of Pol II-bound p53 is shown in Fig. 1d. Both TAD domains are located at a distant position relative to the core domain, connected with an intrinsically disordered loop. Due to the inherent unstructured nature of the proline-rich domain[29], there were insufficient extra cryo-EM densities of p53 that could be assigned to represent this intact domain (Fig. 1b). Nevertheless, we were able to capture a small portion of prominent extra density that is positioned in between Rpb2's protrusion and lobe within Pol II. Thus, we constructed the structure of a partial proline-rich domain (spanning 88 to 92 aa) as a loop that extends from the core domain (Fig. 1d, highlighted in green and Supplementary Fig. S3B). The remaining proline-rich domain (i.e., 62–88 aa) has retained its unstructured property since the cryo-EM density was mostly averaged out during data processing. To accommodate the conjunction of the proline-rich domain and TAD1/2, the region (i.e., 94–109 aa) of the core domain adapts a conformational change that dispossesses a helix and a β-sheet present in a previous crystal structure (PDB: RCSB 3TS8)[34] (Fig. 1d, highlighted in blue and Supplementary Fig. S3A). Furthermore, it was difficult to define the structure of p53's C-terminal domain due to its inherent high plasticity, although sparse extra p53 EM densities were observed close to the oligomerization domain (Fig. 1b). Taken together, these results reveal that p53's functional domains specifically target distinct surfaces within Pol II, indicating the multiplex ability of p53 to structurally regulate Pol II's transcriptional activities.

**The interface between Pol II and the p53 core domain.** The p53 core domain exhibits sequence-specific DNA-binding activity that binds p53 REs[29]. Importantly, approximately 90% of p53 oncogenic mutations occur in the core domain that disrupt its DNA binding function and drive tumorigenesis in variety of human cancer[29]. Hence, we sought to further uncover the molecular basis of how the p53 core domain engages Pol II. Upon close examination of their interaction, several distinct structural features were identified (Fig. 2). First, the contact surface of the p53 core domain/Pol II specifically involves the Rpb1 clamp and the Rpb2 protrusion. Intriguingly, a cavity between p53 core domain and Pol II was formed upon this association (Fig. 2a). The molecular structure of p53/Pol II further reveals that this cavity is positioned between the key DNA-binding helix of p53 and the Rbp1 clamp (Fig. 2b). Notably, several critical DNA-binding residues of p53 (e.g., R175, R248 and R282)[9,29] and Pol II (e.g., P52 and R291)[39], frequently mutated in diverse human cancers, are exposed to the surface around this cavity. This raises the possibility that the cavity may allow the p53 core domain to recognize and bind p53 RE DNA. It may also imply a unique position within p53-bound Pol II to engage target gene promoters.

Another key notion is the select location of the p53 core domain relative to Pol II/DNA interaction. A number of reports have documented how Pol II structurally binds DNA during transcription[6,40,41]. The DNA binding cleft of Pol II, composed of the Rpb1 clamp and the Rpb2 protrusion, is responsible for upstream DNA binding in the absence of p53[6] (Fig. 2c, left panel). The structure shows that the p53 core domain occupies the surface where the DNA-binding cleft of Pol II locates (compare both panels). This suggests that p53 could directly affect DNA binding activity of Pol II, such as preventing Pol II to engage non-specific DNA. In addition, the major part of the p53 core domain comprising the β-sheet sandwich scaffold binds the key regulatory region of Pol II (right panel), which interacts with TFIIA, TFIIB and the TBP subunit of TFIID during the PIC assembly process[6] (left panel). Aside from the above select GTFs, the elongation factor DSIF also binds the same surface of Pol II as the p53 core domain[42], implying the key location within Pol II for regulation of transcriptional activities. Collectively, these results indicate the unique positioning of the core domain, which could allow p53 to directly regulate Pol II's DNA binding as well as recruitment of select GTFs and/or elongation factors for transcription.

**The interaction of the p53 N-terminus with Pol II.** It is established that the N-terminal transactivation domains of p53 interact with diverse transcriptional co-factors (e.g., CBP/p300, TAF6/9 and TFIIH/p62) to activate transcription of vast gene networks that dictate cell fate[13]. Therefore, it would be intriguing to reveal in detail how the N-terminus of p53 structurally interacts with Pol II. Our structure demonstrates that the p53 N-terminal domains, harboring TAD1/2 and the proline-rich domain, mainly bind to the Rpb1, Rpb2, and Rpb5 subunits (Fig. 1). Previous structural studies have showed that the intrinsically flexible TADs of p53 can be mutually stabilized to form helices via its interacting factors (e.g., MDM2[25], replication protein A[26] and CBP/p300[27]). Therefore, based upon a previous NMR structure[27] and the p53 cryo-EM density of our 3D

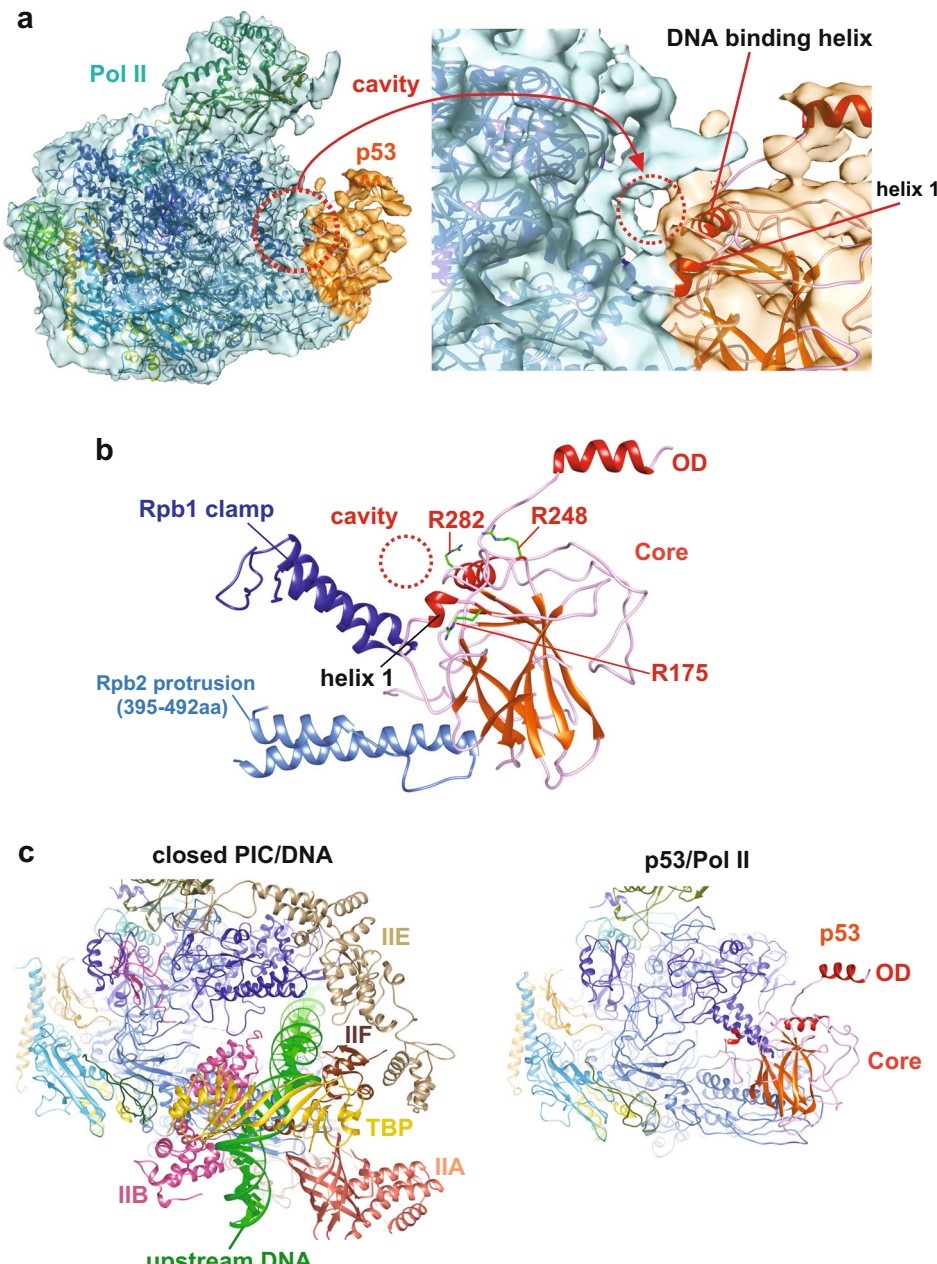

**Fig. 2 Structural features of the p53 core domain when bound to Pol II. a** An overlay of cryo-EM density and p53/Pol II molecular structure is shown (left panel). A zoomed view highlights the p53 core domain and Pol II interface (right panel). The p53 core domain/Pol II cavity (red circle) is formed upon their interaction. The key DNA binding helix and helix 1 of the p53 core domain are indicated. **b** The molecular structure of the p53 core domain and Pol II around the interface is presented. The cavity is positioned in between Pol II's Rpb1 clamp and DNA binding helices of the p53 core domain. Three hot-spot oncogenic residues (i.e., R175, R248 and R282, frequently mutated in cancer patients)[9,29], critical for sequence-specific DNA binding, are located around the surface of the core domain/Pol II cavity. **c** The structure of the closed PIC (PDB: RCSB 5IYA[6], left panel) shows the positioning of Pol II and upstream DNA/GTFs, in comparison with the contact surfaces of p53-bound Pol II (right panel). The p53 core domain occupies the regions of Pol II in the vicinity of upstream DNA, TBP, TFIIA and TFIIB (compare both panels).

reconstruction, we have identified the formation of the TAD1/2 helices that bridge the Rpb1 jaw and the Rpb5 jaw (Fig. 3a). To obtain a better structural assignment and prediction, partial TADs (i.e., 28–45 aa) were further de novo constructed via Rosetta[43]. Moreover, the de novo structure of p53 residues 1–11 was difficult to solve, as different reconstructions lead to inconsistent results. In addition, our 3D reconstruction of p53/Pol II preserves a small Pol II cryo-EM density extending from Rpb1 between TAD1 and Rpb5, which was previously unresolved in known Pol II structures[6]. We assigned this density to represent

the loop of Rpb1's residues 158–179 via de novo construction (Fig. 3a), which is consistent with a recent MD simulation study of the entire PIC assembly[4].

Formation of TAD1's two helices (spanning 12-25 aa and 26-43 aa, respectively) was aided by Rpb1 (150–218 aa) and Rpb5 dual helices. The TAD2 helix appeared adjacent to the Rpb1 jaw (Fig. 3a). Both regions of Rpb1 (i.e., residues 150–180 and 181–218) could be mutually stabilized via TAD1/2. The individual helices of the TAD1/2 domain bound to Pol II are structurally conserved with TAD1/2 helices bound to two

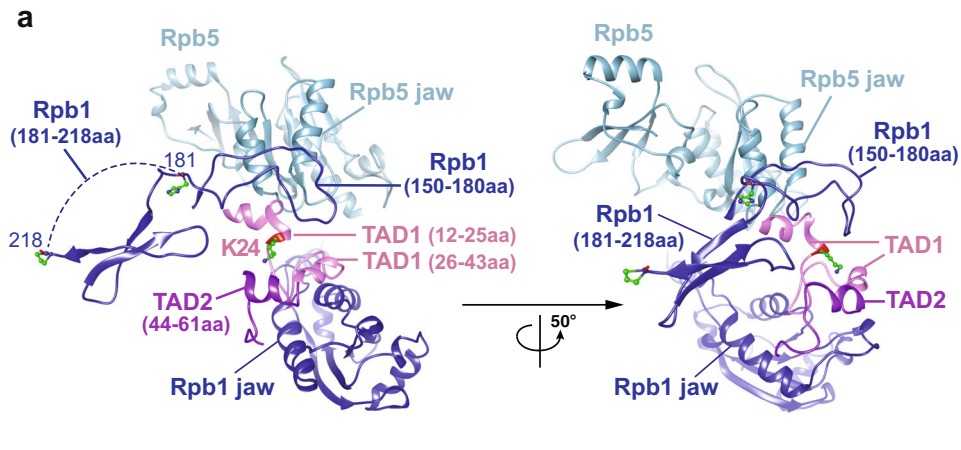

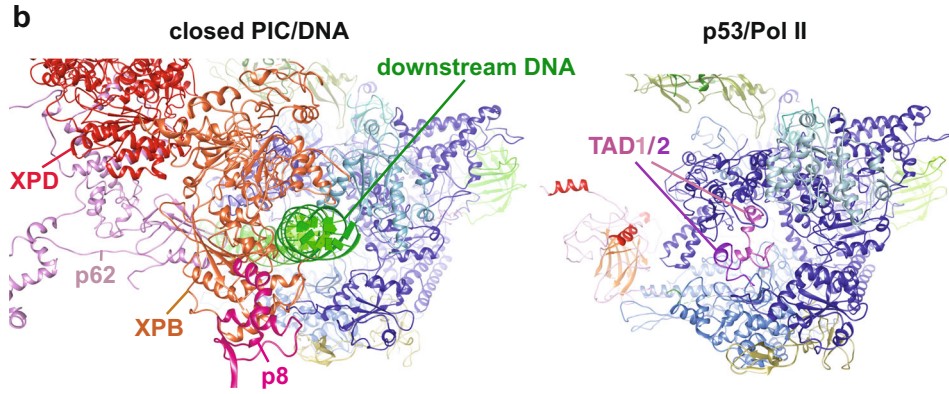

**Fig. 3 Positioning of the p53 N-terminal TADs within Pol II. a** Two views of p53's N-terminal transactivation domains (i.e., TAD1/2) with surrounding Pol II subunits are shown. TAD1 is mutually stabilized by the Rpb5 helical bundle (Rpb5 jaw) and Rpb1 residues 150-218. TAD2 is structurally stabilized by the Rpb1 jaw. **b** A corresponding perspective view of the closed PIC (PDB: RCSB 6O9L[4], left panel) that displays Pol II and downstream DNA/GTFs is compared with the TADs' position within the p53/Pol II co-complex (right panel). TAD1/2 occupy similar surfaces as downstream DNA, adjacent to XPB, XPD and p8 of TFIIH. For clarity, TBP, TFIIA, TFIIB, TFIIE, TFIIF and select TFIIH subunits (i.e., p52, p44, p34 & MAT1) are omitted.

different regions of CBP[27,44] (Supplementary Fig. S4B). However, the orientations between the TAD1/2 helices within this domain are structurally heterogenous amongst all structures (Supplementary Fig. S4C) likely indicating an interaction specific co-folding of the TAD1/2 domain. This structural assignment of TADs within Pol II is in agreement with our previous biochemical studies[22], showing that the p53 N-terminus specifically interacted with the Rpb1 subunit within the intact Pol II. Notably, the select TADs location within Pol II coincide with downstream DNA and TFIIH binding sites[4] (Fig. 3b, compare both panels). Taken together, these observations demonstrate the unique positioning of p53's N-terminal TADs that anchor the Rpb1/5 jaw, indicative of a novel accessibility for p53 to interact with additional transcriptional co-factors (e.g., select GTFs and Mediator[18]) when bound to Pol II.

**Conformational changes within Pol II upon p53 binding.** A battery of evidence has shown that Pol II is not structurally static, exhibiting local conformational changes upon binding to DNA and other interacting factors[5]. Indeed, we observed that several Pol II subdomains underwent prominent conformational changes upon p53 binding (Fig. 4). First, we found that the Pol II clamp coiled-coil shifted further downwards. To verify this distinct structural state, we further performed multi-body refinement analysis on the 3D reconstruction of p53/Pol II via RELION[45] to

improve the cryo-EM density harboring the Pol II clamp coiled-coil region (Fig. 4a and Supplementary Fig. S5).

Following the validation of our clamp structural status, we next compared it with the clamp position of the closed Pol II structure (present in the closed PIC)[6]. Intriguingly, our Pol II clamp coiled-coil moves ~30° downward near the p53 β-sheet sandwich scaffold and the Rpb2 protrusion (Fig. 4b). This local movement has thereby led to a further-closed structural state of the Pol II clamp coiled-coil within the p53/Pol II assembly. In addition, the interaction between Pol II and the p53 core domain/partial proline-rich domain introduced a ~7° rotational shift of the Rpb2 protrusion (Fig. 4b).

Additional structural changes of Pol II upon p53 binding were identified when comparing with the closed Pol II[6]. The Rpb2 lobe of our Pol II rotates approximately 2° downward when engaging and likely to structurally supporting the proline-rich domain (Fig. 4c, highlight in green). Moreover, the Pol II stalk (composed of Rpb4 and Rpb7) shifts ~7°–8° towards the p53 core domain/oligomerization domain with no direct contacts (Fig. 4d). We surmise that the Pol II stalk movement within p53/Pol II may consequently correspond to the further-closed structural change of the Pol II clamp coiled-coil. Overall, these findings reveal key local conformational changes of Pol II induced by p53 binding. The identified Pol II clamp coiled-coil position within p53/Pol II indicates that the p53 core domain could directly modulate Pol II DNA binding activity (Figs. 2c and 4).

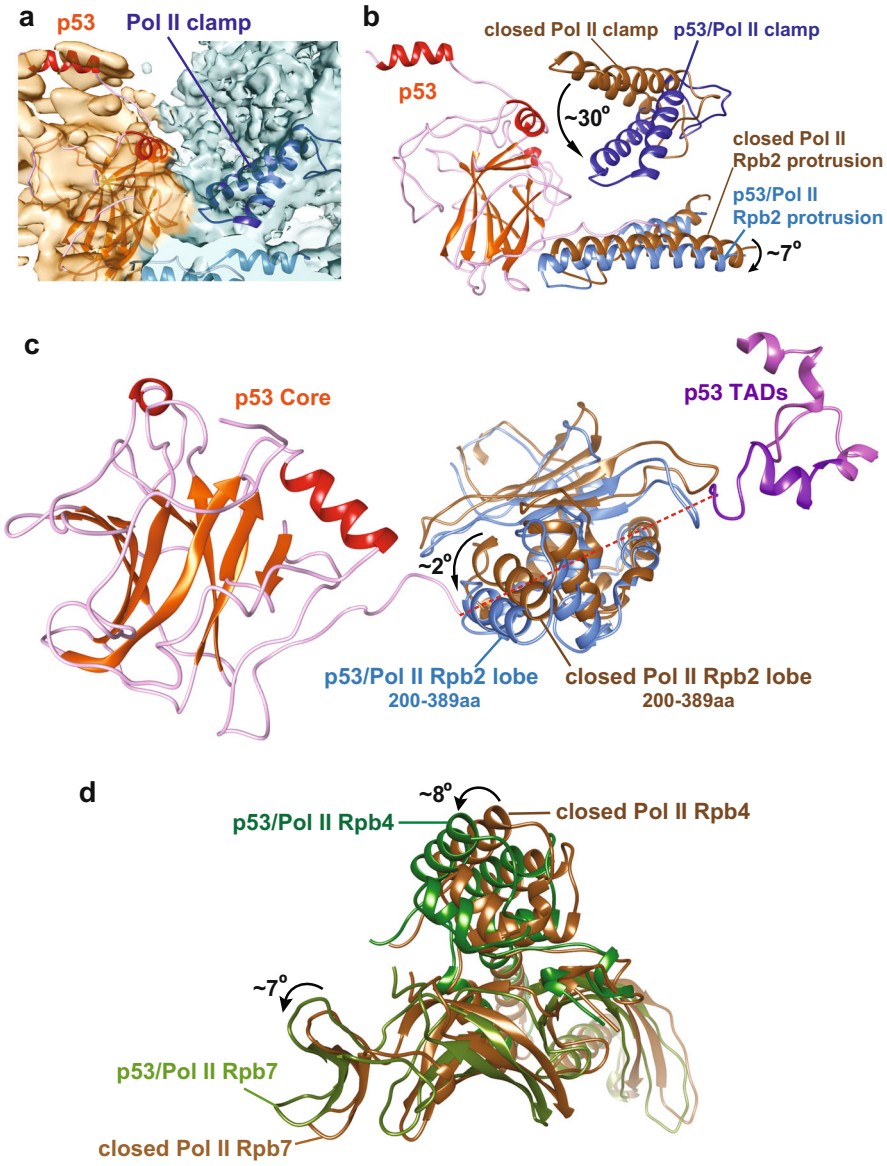

**Fig. 4 Conformational changes of Pol II upon p53 binding. a** Multi-body refinement of the p53/Pol II 3D reconstruction was performed to improve the cryo-EM density covering the Pol II clamp coiled-coil region (Pol II in teal color and p53 in orange color). Pol II clamp displaying a signature feature of the coiled-coil (dark blue) is indicated. **b** Upon p53 core domain binding, conformational changes of Pol II clamp coiled-coil and Rpb2 protrusion within Pol II are observed via comparing both molecular structures of closed Pol II (PDB: RCSB 5IYA[6], brown) and p53-bound Pol II (dark blue). p53 binding introduces movement of Pol II clamp coiled-coil, approximately ~30° downwards into a further-closed state, along with ~7° downward movement of Rpb2 protrusion (sky blue). **c** The Rpb2 lobe of p53-bound Pol II (sky blue) moves ~2° downwards compared to the closed Pol II (PDB: RCSB 5IYA[6], brown). This structural change of the Rpb2 lobe would likely be induced by the proline-rich domain of p53 (indicated as green line and the red dash line). **d** p53-bound Pol II stalk (composed of Rpb4/7, green) is in comparison with the closed Pol II stalk (PDB: RCSB 5IYA[6], brown). Distinct conformational change of the Pol II stalk upon p53 interaction is observed, since Rpb4 and Rpb7 within p53/Pol II move approximately 8° and 1Å, and 7° and 3 Å, respectively.

## Discussion

**Functional domains of p53 target key regulatory surfaces within Pol II.** Recent advances in single particle cryo-EM permit the high-resolution structural determination of RNA polymerase II (Pol II) in various phases of the transcription cycle[40,42,46]. In addition, the transcription initiation process has been elucidated by structural studies of Pol II in complex with other general transcription factors[4] and Mediator[2]. Thus far, high-resolution tertiary structures of human Pol II bound to transcriptional activators have been technically challenging, largely due to structural heterogeneity and inherent plasticity. By overcoming a number of technical hurdles, we have finally uncovered a high-resolution 3D structure of the tumor suppressor p53 bound to Pol

II, in order to gain structural insights of how p53 may directly regulate Pol II's activities.

Our p53/Pol II structure reveals that p53's main functional domains specifically target key regulatory regions within Pol II. First, the p53 core domain binds the DNA binding cleft of Pol II where upstream DNA, the Rpb1 clamp and the Rpb2 protrusion are located (Figs. 1 and 2). Notably, this interaction introduces movement of the Pol II clamp coiled-coil closer towards the Rpb2 protrusion (Fig. 4a, b). Previous studies have shown that the open-state coiled-coil permits loading of DNA into the Pol II cleft and subsequently adopts a closed state to seal the cleft during elongation[47]. This suggests a critical role of the coiled-coil in controlling the open/closed state of the Pol II clamp that engages

DNA. Furthermore, a closed-state coiled-coil was observed in both Archaeal Pol II (PDB: RCSB 2PMZ)[48] and GTFs/DNA-bound Pol II (closed PIC, PDB: RCSB 5IYA)[6]. The distance between both tips of the Rpb1 clamp coiled-coil and the Rpb2 protrusion is approximately 45 Å, as measured in both Archaeal Pol II (Rpo1'/A257 relative to Rpo2/K373) and human Pol II (Rpb1/N296 relative to Rpb2/D424) within the closed PIC. Intriguingly, the distance in our p53/Pol II structure (Rpb1/N296 relative to Rpb2/D424) is approximately 17 Å, which is much shorter than the distance (i.e., ~45 Å) in both Archaeal Pol II and human Pol II (closed PIC). This observation suggests that, in the absence of DNA, the p53-bound Pol II clamp adopts a further-closed conformational state. This further-closed state would likely only be possible when not bound to DNA as this clamp conformation is incompatible in an elongating Pol II due to steric clashes with DNA in Pol II's cleft[40,49]. We speculate that p53 and the further-closed clamp may act as a gate to hinder DNA's access to Pol II's cleft thereby mitigating non-specific binding to DNA. To the best of our knowledge, this finding implies that p53 introduces a new structural feature of the Pol II clamp, which could negatively regulate the DNA binding activity of Pol II.

It is established that, in a transcribing DNA-bound Pol II, the SPT4/5 subunits of DSIF structurally modulate the position of the Pol II clamp in open/closed transitions[41,47,50–52]. Select domains of DSIF (i.e., Spt5-NGN/Spt4 and Kow1-L1) serve as a DNA clamp to stabilize the upstream DNA-binding channel[49]. It is of note that the p53 core domain binds the same key surface within Pol II as the elongation factor DSIF[42,49,51]. Importantly, the Pol II clamp movement induced by p53 also occurs with DSIF-bound Pol II in the transcribing state[49]. Based upon the DSIF-bound elongation complex (PDB: RCSB 5OIK)[49], the distance between both tips of the Rpb1 clamp coiled-coil and Rpb2 protrusion (Rpb1/N296 relative to Rpb2/D424) is ~36 Å, which is 9 Å shorter than human Pol II in the closed PIC assembly. Our previous studies have shown that the interaction of p53 and Pol II directly enhances transcription elongation efficiency of Pol II[22]. We thereby surmise that, when Pol II embarks DNA in the transcribing state, the p53 core domain may mimic SPT5 of DSIF to structurally stabilize DNA[22]. In an elongating Pol II, we speculate that p53 may induce a similar Pol II clamp conformation present in the DSIF/Pol II assembly since a further-closed clamp state would be predicted to sterically clash with the DNA. Collectively, these findings suggest that p53 regulates Pol II's DNA binding activity to mediate transcription initiation and elongation processes via multiple mechanisms.

One important observation in this study is the distinct position of the N-terminal transactivation domains of p53 (i.e., TAD1/2), which wasn't yet resolved in our previous lower resolution cryo-EM structure of p53/Pol II[22]. Our structure reveals that TAD1/2 are located in between the Pol II jaw (composed of Rpb1 and Rpb5) while distal to the core domain/C-terminus of p53 (Figs. 1 and 3). We have determined that structurally flexible TADs were mutually stabilized by Rpb1 and Rpb5 (Fig. 3a). Importantly, the p53 N-terminus is known to interact with its regulatory factors including TFIID[17], TFIIH[53], hdm2/mdmX[54] and p300/CBP[55]. Thus, the distinct TAD position distal to core domain/C-terminus within Pol II may imply the possibility for p53 to recruit additional co-factors when binding Pol II to regulate transcription.

Notably, the structure of p53/Pol II shows that TADs specifically occupy the region of Pol II where downstream DNA traverses[4,6] (Fig. 3b). A previous NMR study reported that TAD2's helices could structurally mimic single strand DNA when bound to replication protein A[26,56], probably due to the amphipathic nature of TAD2. We thereby speculate that the location of p53 TADs at the Pol II jaw could impact Pol II's interaction with downstream DNA. In support of this concept, the p53 N-terminus specifically engages the DNA-binding region of its core domain to reduce non-specific DNA binding[56]. The distant location of p53's TADs relative to the core domain within Pol II may serve to release the core domain for binding target consensus DNA (i.e., p53 REs). Therefore, the select positioning of TADs within Pol II poses a novel mechanism for p53 to mediate Pol II's access to target DNA.

**Potential novel recruitment directed by the p53 core domain/C-terminus.** It is well documented that transcriptional activators, including p53, stimulate Pol II recruitment and PIC formation to activate gene expression[9,57]. However, the precise molecular mechanism underlying how p53 directly regulates Pol II's activities remains unclear. For example, p53-mediated Pol II recruitment on DNA could be driven by p53's ability to bind DNA and/or its interactions with other PIC components (e.g., TFIID and Mediator)[17,18]. The central core domain allows p53 to recognize and bind consensus DNA sequences (i.e., response elements, REs) within target genes. Intriguingly, the structural framework of p53/Pol II creates a cavity located in the distinct interface between the Rpb1 clamp and p53's key DNA-binding helix of the core domain (Fig. 2). It appears that critical residues responsible for p53 DNA binding[58] are not embedded in contact surfaces, but exposed to the p53 core domain/Pol II cavity instead (Fig. 2b). This finding poses the possibility that, upon contacting Pol II, the p53 core domain could retain its ability for RE recognition[29]. Based upon the dimension of this cavity (~ 17Å diameter measured), we suspect that it might be possible for the p53 core domain to engage sequence-specific DNA containing p53 REs, and/or even lock DNA within the cavity[59]. More importantly, this p53 core domain/Pol II cavity is surrounded by several crucial residues of p53 (e.g., R175, R248 and R282)[9,29] and Pol II (e.g., P52 and R291)[39] that are frequently mutated in various human cancers, further highlighting the potential biological significance of the cavity.

To further support our hypothesis regarding the cavity's capability to host DNA, we performed several molecular dynamics (MD) simulations on DNA-bound p53/Pol II assembly. With three different lengths of DNA segments tested, stable p53/Pol II/DNA structures with negative free binding energies were obtained via MD simulations. The most frequently appearing model of p53/Pol II/DNA is shown in Fig. 5. The MD model reveals that the DNA could traverse through the p53-Pol II cavity towards Rpb 1/5 jaws. The DNA binding helix (specifically the residue R282) of p53 was indeed involved in association with the upstream DNA, consistent with previous reports on the p53 core domain/DNA interaction[29,34]. It is known that oligomeric p53 binds target genes[59]. However, despite 25 years of intensive studies, it remains unclear whether p53 must form as an oligomer (i.e., dimer/tetramer), in order to associate with its co-factors and bind its RE DNA[60]. Our MD model suggests that Pol II-bound monomeric p53 could engage DNA via the cavity without the formation of a tetramer or dimer. Future cryo-EM work on DNA-bound p53/Pol II will further validate this observation. Overall, the unique positioning of the p53 core domain within Pol II may provide both a geometric capability for RE recognition and a novel position for Pol II to interact with target p53 gene promoters.

**Monomeric p53 bound to Pol II.** We would like to point out that our 3D reconstructions and the molecular structure of the p53/Pol II assembly can only be assigned and represented as monomeric p53 bound to Pol II (both in this study and our previous work[22]). Importantly, we cannot rule out that an oligomeric form

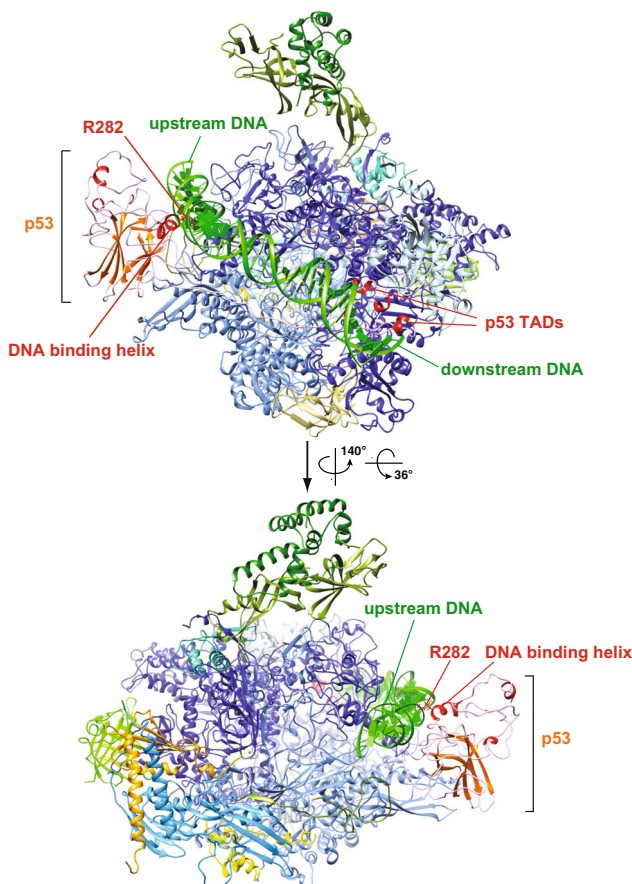

**Fig. 5 Proposed model of the p53/Pol II cavity targeting DNA.** A proposed model of the p53/Pol II cavity that interacts with DNA was generated via molecular dynamics simulation. The structure most frequently appearing during 20ns production runs is presented. The key DNA binding residue of p53, R282, is indicated.

of p53 (i.e., dimers/tetramers) can associate with Pol II. First, given that p53's oligomerization domain is structurally exposed on top of the central core domain and not in contact with Pol II (Fig. 1), it seems feasible for the Pol II-bound monomeric p53 to recruit additional monomers and further aid DNA binding. In addition, our previous 2D classification analysis identified that p53 could mediate Pol II dimerization while no Pol II dimers were detected without p53[61]. These results indicate that Pol II-bound monomeric p53 may hypothetically form dimers to direct Pol II dimerization and facilitate their occupancies on target gene promoters containing pre-bound dimeric p53 (Supplementary Fig. S6). In such a model, a conformational change in the p53/Pol II interface would be necessary to allow the loading of DNA into Pol II's cleft during PIC formation (Supplementary Fig. S6). Along with this notion, Pol II clustering around gene loci was identified via single molecule studies[62,63]. This further implies that p53 could load, at least, two Pol II molecules at a time via oligomerization to stimulate transcription bursting. Nevertheless, it remains an open question whether oligomeric p53 interacts with Pol II in the absence or presence of target gene DNA.

**The p53 core domain/TADs may be essential to promote early-state PIC assembly.** It is well established that p53 promotes PIC formation via, at least in part, its interactions with other PIC components such as TFIID, TFIIH, and Mediator[17,18,53]. A number of studies have shown that p53 escorts GTFs to recognize

promoter DNA and facilitate transcription[23,53,64]. Our previous docking analysis[6,22,61] (with superimposed cryo-EM p53/Pol II structure into the closed PIC) indicates the possibility of p53's co-presence during PIC assembly. This event may occur, because p53's mobility could help adapt its position without structural clashes when recruiting other PIC components such as TFIIA or TBP/TFIID[23]. Perhaps upon association with Pol II, p53 could temporarily lock DNA in the cavity while recruiting other components of GTFs to directly facilitate PIC formation.

Earlier reports have documented that, independent of PIC formation, p53's TADs can interact with the p62 subunit of TFIIH to regulate p53 target gene expression[16,53,65]. Moreover, both TFIIE and p53's TADs share a similar binding site within p62 of TFIIH[15]. Notably, our molecular structure (Fig. 3b) shows that Pol II-bound TADs are positioned in close proximity to the binding site of the XPB, XPD and p8 subunits of TFIIH within the PIC[4,6]. This select location suggests that the N-terminal TADs of p53 would not compete with TFIIE for binding p62 when in complex with the PIC (Figs. 2 and 3). To complete the entire PIC assembly process, we suspect that p53 may shift its position to permit sequential recruitment of other GTFs, including TFIIB and TFIIH. Consistent with our previous studies[23], the structural compatibility of p53/Pol II with several key components of GTFs suggests p53's regulatory role in stimulating the early stage of PIC assembly. Overall, our findings suggest a multiplex mechanism by p53 to structurally regulate Pol II's activities in engaging DNA and additional factors (e.g., GTFs and p53 monomers), thereby providing further insights into p53-mediated Pol II transcriptional activation.

## Methods

**Protein purification and cryo-EM sample grid preparation.** The purification of the p53/Pol II co-complex and the subsequent cryo-EM sample grid preparation were performed as described in the Materials and Methods section of our previous report[22], except that 32 μg of endogenous hdm2 DNA fragment was added during incubation and the peptide recognized by Pol II monoclonal antibody 8WG16 for protein elution was purchased from GenScript. The Pol II monoclonal antibody (8WG16) can be obtained from Sigma–Aldrich (cat. no. 05-952-I). HeLa cells can be obtained from ATCC (ATCC CCL-2).

**Cryo-EM and single particle 3D reconstruction of p53/Pol II.** Cryo-EM data was collected with a FEI Titan Krios cryo-EM microscope (located at HHMI Janelia Research Campus) operated at 300kV with a defocused range −0.5 μm to −2.5 μm. All digital micrographs were obtained using a K2 summit direct electron detector (Gatan) operating in super-resolution mode at magnification of 38,168 (0.655 Å/pixel). Fifty-frame exposures were taken at 0.2 second per frame (10 seconds total exposure time) and a total dose of 58 electrons per square angstrom per micrograph.

Approximately 5,500 movies were acquired and further binned by 2-fold to 1.31 Å/pixel for faster data storage, and drift-corrected by MotionCor2[66] (Supplementary Fig. S1A). Contrast transfer function parameters were calculated by Gctf[67]. Poor-quality and low-resolution micrographs were removed prior to particle analysis and 3D reconstruction. After a 10 Å low-pass filter was applied to all micrographs, a total of 776,710 particles were automatically picked by SPHIRE-crYOLO[68] along with PhosaurusNet network. Selected particles were extracted and further binned (2.62 Å/pixel) using RELION-3.0[69]. Two rounds of reference-free 2D classification analysis were conducted to remove junk/contaminants and collect non-DNA bound p53/Pol II particles for this structural study.

A total of 682,938 particles were used for unsupervised 3D classification analysis using a 60 Å low-pass filtered initial model of p53/Pol II. This initial reference model was independently generated from the particle dataset (105,390 particles, using the same set of micrographs), which were exclusively picked/processed by RELION-3.0[69] and produced an 8 Å resolution 3D reconstruction (see Supplementary Methods). Initial 3D classification analysis (T=4 and 7.5° sampling) generated 8 different classes (Supplementary Fig. S1B). Particles from two prominent 3D classes with similar views of the tilted Pol II stalk were pooled and subjected to another round of 3D classification with identical parameters as above. Particles from those two classes were re-extracted and re-scaled to 1.31 Å/pixel, followed by another round of 3D classification with one final class. The resulting 3D class volume (low-pass filtered to 50 Å) was applied as an initial reference map for subsequent 3D auto-refinement analysis (Supplementary Fig. S1B). The initial 3D refinement was carried out with a soften mask generated from the initial reference 3D volume with the following parameters: 50 Å low-pass

filter, 10 pixels extension and 5 pixels for soften edge. The 3D reconstruction of p53/Pol II was obtained at a global resolution of 4.4 Å, corresponding to the gold standard Fourier Shell Correlation using the 0.143 criterion (Supplementary Fig. S2).

To further improve the cryo-EM density of the p53 core domain within Pol II, we next performed focused 3D classification with a mask covering the p53 core domain. Focused 3D classification ($T=20$) was conducted without particle re-alignment and generated two different classes (Supplementary Fig. S1B). One of the classes with improved cryo-EM density of the p53 core domain was subsequently proceeded with 3D refinement and post-processing. The final structure of p53/Pol II was generated at a global resolution of 4.6 Å based upon FSC curve at 0.143 value (Supplementary Figs. S1 and S2A and Supplementary Movie 1). Euler angle distribution of particle views used in the reconstruction was determined using RELION and its 3D plot was visualized in Chimera (Supplementary Fig. S2B and Supplementary Movie 2). Local resolution was estimated by Resmap[70] (Supplementary Fig. S2C). Figures were generated using UCSF Chimera[71].

**Model fitting and de novo building**. Human Pol II structure was adapted from the closed-form pre-initiation complex (PIC, PDB: RCSB 5IYA)[6]. The core domain and N-terminus of human p53 were based upon RCSB 3TS8[34] and 2L14[27], respectively. Specifically, mutagenesis and deletions of human p53 protein (i.e., RCSB 3TS8[34] and 2L14[27]) were restored to wild-type p53 sequences (refer to UniProtKB P04637) prior to the structural assignment. The initial molecular structures of Pol II and p53 core domain (residues 94-291) were first aligned by Phenix[37] with manually adjusted clamp domains of Pol II. We implemented COOT[72] to adjust residues 324-356 and residues 94-109 of p53, and manually build residues 83-94 and 292-323 of p53. To best construct the structure of the N-terminus p53, TAD1 (residues 13-27) was aligned to mimic the NMR structure (PDB: RCSB 2L14)[27], which is located underneath the double helix bundle of Rpb5. Residues 28-45 of p53 were de novo built by Rosetta[43,73] for better structure alignment in this highly mobile region. Finally, residues 158-179 of Rpb1 were de novo built by Rosetta[43,73] to better fit the cryo-EM density of p53/Pol II. The consensus structure docking and refinement of p53/Pol II complex was achieved by Phenix[37].

**Multi-body refinement of p53/Pol II**. To better define the Rpb1 clamp coiled-coil cryo-EM density, multi-body refinement was performed via RELION as described in a previous report[45]. In this refinement, a lone Pol II volume was generated based upon the 6 Å resolution 3D map of human closed Pol II form (PDB: RCSB 5IYA)[6], and then was used to subtract p53/Pol II density to produce a lone p53 volume. Pol II and p53 masks were subsequently generated by a "relion_mask_create" function, with an extended 5 pixels of binarized maps, 5 pixels of soft edge and 10 Å of low-pass filter applied to the original lone Pol II and p53 volumes.

**Molecular dynamics simulation**. To generate an initial model of DNA-bound p53/Pol II complex for molecular dynamics (MD) simulation analysis, a DNA segment of the *hdm2* gene promoter was incorporated into the p53/Pol II cryo-EM structure obtained from this report. The initial model was constructed by TLeap module of AMBER 16[74]. We adapted ff14SB[75] and OL15[76] for protein and DNA forcefields, respectively. The minimum distance between the surface atom and the periodic boundary is 12 Å, solvated with TIP3P water. Potassium counterions and an additional 150 mM NaCl were added to balance the charge and mimic physiological conditions, respectively.

The system was energy minimized for 8000 steps with fixation of every atom, followed by another 8000 steps with a harmonic restraint on heavy atoms ($k = 100$ kcal mol$^{-1}$ Å$^{-2}$). A 400,000-step annealing brings the system from 0K to 300K over 500ps, with a Langevin thermostat with 10.0 ps$^{-1}$ collision frequency. All harmonic restraints were gradually released in a seven-step equilibrium run (total 270 ps). Production runs were performed in the NPT ensemble (1 atm and 300 K) for 10ns, employing periodic condition. The non-bonded interaction cut-off was 10.0 Å, and the hydrogen bonds were constrained by the SHAKE algorithm[77]. Data analysis of the MD trajectories was performed by CPPTRAJ[78]. The RMSD calculation is based on the initial model as the reference. Cluster analysis was performed using the average-linkage method, and coordinate RMSD was the distance metric. The analysis stops when either five clusters are reached, or the minimum distance between clusters is 3.0 Å.

**Statistics and reproducibility**. Cryo-EM microscopy yielded 5500 micrographs containing approximately 776,710 particles. The micrographs with bad ice were excluded as is common practice in cryo-EM field. An earlier study[22] with a different particle set results in ~11 Å resolution density map, which is similar as current results. In addition, two different approaches of particle picking were used which yielded similar density maps but with different resolutions. Randomization in the data analysis was achieved via unbiased particle classification (e.g., maximum likelihood) in RELION.

**Reporting summary**. Further information on research design is available in the Nature Research Reporting Summary linked to this article.

## Data availability
Data supporting the findings of this manuscript are available from the corresponding author upon request. Cryo-EM 3D reconstruction of p53/Pol II and its molecular structure have been uploaded to the Electronic Microscopy Data Bank (EMD: 22294) and Protein Data Bank (PDB: 6XRE), respectively.

## Code availability
No custom code was used in this manuscript.

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

## Acknowledgements

We thank S. Zheng and R. Tjian for providing Pol II mAb supernatant. We appreciate assistance from the AIF facility at Albert Einstein College of Medicine (Einstein), especially F. P. Macaluso and L. Cummins. We specially thank B. Carragher, P. Clint, E. Eng and W. Rice for helpful suggestions, cryo-EM sample grid screening and initial image acquisition at the New York Structural Biology Center. We are grateful to Z. Yu and C. Hong at HHMI Janelia Research Campus cryo-EM facilities for help in microscope operation and high-throughput data collection. We also thank F. DiMaio, R.Y. Wang and P. Adonine for helpful discussion regarding structural assignment. We appreciate L. Wang at University of California at Davis for permission to access MPS computing cluster for MD simulation. This work was supported by Einstein start-up fund and NIH/NIGMS grant (1R01GM126045-03 to R. A. Coleman). Initial work was performed at the Simons Electron Microscopy Center and National Resource for Automated Molecular Microscopy located at the New York Structural Biology Center,

supported by grants from the Simons Foundation (SF349247), NYSTAR, and the NIH National Institute of General Medical Sciences (GM103310) with additional support from Agouron Institute (F00316) and NIH (OD019994) for Krios1 microscope. Molecular graphics and analyses were performed using UCSF Chimera at the University of California, San Francisco supported by NIH P41-GM103311. W. Liu is an affiliated member of the New York Structural Biology Center.

## Author contributions

W.L. designed the experiments and developed the project. R.A.C. performed sample preparation and provided suggestions. S.K.S. prepared sample grids and collected data. R. H.S. assisted data collection at HHMI Janelia Research Campus. S.-H.L. performed single particle 3D reconstruction, structural assignment and MD simulation. S.-H.L., W.L. and R.A.C. interpreted the data and wrote manuscript. W.L. and R.A.C. supervised experiments and data analysis.

## Competing interests

The authors declare no competing interests.
