## [Peer Review File · Communications Biology]

Reviewers' comments:

Reviewer #1 (Remarks to the Author):

In this manuscript, Wei-Li Liu and colleagues use single-particle cryo-EM in combination with computational modeling to deduce the structure of the human p53/Pol II complex to 4.6 Å resolution. The study is a follow-up of the authors' G&D 2016 paper (ref. 22) that determined 11 Å resolution cryo-EM structure of the same complex. The principal advances reported here consist of detection of conformational changes in the Pol II clamp coiled-coil, presence of a 17 Å wide cavity between the DNA-binding surface of p53 and Pol II, and localization of the TADs in between the Pol II jaw contributed by the Rpb1 and Rpb5 subunits. While these additional insights into p53/Pol II structure are not substantial, they do clarify how p53 might stimulate PIC formation. The paper has outstanding graphics; the molecular structures go well beyond those presented in their previous publication.

Principal concerns

1. If p53 binding induces movement of Pol II clamp into a further-closed state, it's unclear that the interaction between p53 and Pol II will enhance the efficiency of Pol II elongation as claimed. Is there any evidence of co-localization of p53 and Pol II within transcribed regions in mammalian cells? More thoughtful discussion on this point is required.
2. An unusual aspect of this and the precursor study is the implied mechanism of p53-Pol II association, namely, that it's the DNA-free p53 protein (possibly monomeric) that initially binds Pol II and that this complex subsequently associates with DNA-bound p53. This hypothetical mechanism, which to my knowledge has no precedent, needs to be made clearer. A schematic would be helpful.

Minor points

1. Manuscript was written as if the reader had just finished reading the authors' G&D 2016 paper. This is unlikely to be the case. Results section requires more explanation on how the p53/Pol II co-complex was assembled, how the figures were generated, among other details.
2. While the ms. was accurately and clearly written for the most part, there are a few places where it was not. For example: p. 3, line 4, "initiation" not elongation; p. 3, second paragraph, line 1, "multiple" not multiplex; p. 4, lines 1-3, entire sentence (awkward construction); p. 8, lines 1-2 (p53 core domain bound to DNA per se does not activate transcription); p. 8, line 12, "the possibility", not a possibility; p. 8, second paragraph, line 1, "...relative to Pol II/DNA" not related to Pol II/DNA; p. 18, second paragraph, line 1, "microscope" not microscopy.
3. Recommend changing "Core" to the more commonly used "core domain" throughout ms.
4. Miscitation: Ref. 38.
5. Figure 5: R282 is not indicated.

Reviewer #2 (Remarks to the Author):

Liou et al. present high resolution structural characterization of the p53-Pol II complex using single particle cryo-EM, revealing new insights into the mechanism of p53-dependent gene regulation. The manuscript is well written and they have done a good job selecting figures that illustrate important features of the structure models. All details leading to the cryo-EM reconstructions, models and fittings are very clearly described. There is simply nothing to be complained about the technical

approaches chosen by the authors. I highly recommend the publication of this nice piece of work.

Reviewer #3 (Remarks to the Author):

In this manuscript Liou et al., describe the cryo-EM structure of an assembly between RNA polymerase II and p53. The authors detail the site of p53 association on RNA Pol II and highlight a set of conformational changes within the Pol II clamp and identify a potential DNA binding cavity between the p53 core and Pol II. Overall the paper is well written, and the insights gleaned from the structure are of broad interest to the field. However, there are a number of inconsistencies that should be addressed and data that need to be visible before this manuscript could be considered appropriate for publication.

MAJOR CONCERNS

The cryo-EM density presented in the manuscript do not convincingly and unambiguously support the atomistic models presented in the Figures. While it is certainly clear that the presented cryo-EM map is markedly improved on previous structures, it is unclear from the figures presented if the cryo-EM density supports the proposed model unambiguously. The authors are strongly encouraged to carefully re-work the figure panels to include clear representations of the EM density to provide evidence to the reader that the maps support the modeled interactions discussed. Specific concerns are broken down into two sections below, firstly some general comments about the overall quality of the reconstruction/model followed by specific concerns regarding the highlighted interactions relevant to the manuscript.

1. General Map Quality

The authors should provide a main text table bearing the data collection, reconstruction, and model statistics. It is difficult to assess the quality of the reconstruction and the fit of the model without reference to the model resolution based on an FSC threshold of 0.5, the Map CC (box), and the Map CC (mask). This table is standard in the field for presenting cryo-EM data and should definitely be included. Furthermore, the authors have no PDB code, suggesting that the provided map/model has not yet been curated by the PDB beyond the preliminary validation. This should be completed and the final validation report provided before further revision of the manuscript.

The Euler angle distribution suggests that the reconstruction suffers from preferred orientation, potentially explaining the streaky nature of the density in certain orientations. Could the authors comment on their confidence in the assigned interactions with Pol II given the limited orientations used for 3D reconstruction?

2. Specific Interactions

A number of regions within p53 are re-modelled based on the cryo-EM map into conformations not previously observed within NMR, X-ray or Cryo-EM structures. Since the EM density is the main source of data in this study, the authors should include supplementary figures where the modified regions of interest are shown within the EM density (semi-transparent) so that the reader can evaluate the confidence of the modeling.

One focus of the manuscript is the presence of a cavity between Pol II and the p53 core domain that is postulated to accommodate double-stranded DNA (Figure 2A). Based on published literature it

appears reasonable that this site could accommodate the p53 response element DNA, especially when looking at panel 2B where there appears to be a substantial cleft that would reasonably accommodate dsDNA. However, looking at the cryo-EM density shown within Figure 2A there are a number of spurious features within the map for which no protein is modelled that appear to occlude the proposed cavity. This is further reinforced in panel 4A where the cavity is not visible at all. Could the authors comment on the large amount of unmodelled density in this region? Are there regions within Pol II or p53 that could be explained by these densities?

Consistent with the above comment, the authors present a remodeled conformation of the Pol II clamp coiled coil which has been modelled 'further downwards'. However, looking at EM density presented in Figure 4A, it does not appear unambiguously convincing that this is appropriate. There are a lot of spurious features in the density that could be attributed to the closed loop conformation of the Pol II clamp. It is also possible that the 3D reconstruction suffers from heterogeneity caused by the Pol II clamp adopting a closed and open conformation. As a result, the map may represent an average of the Pol II clamp in an open and closed state which results in this smearing of the density. Have the authors considered 3D classification in RELION using partial particle subtraction to resolve this? This could both improve the quality of the density and serve to provide higher resolution of the interfaces.

Finally, the N-terminal transactivation domain of p53, TAD1/2, are modelled interacting with the Rpb1, Rpb2, and Rpb5 subunits of RNA Pol II (Figure 3A). This is an instance when the authors should present a detailed cryo-EM map showing the modelled regions within the density so that the reader can evaluate the proposed interaction from the actual data.

Minor Comments

Page 12 – "near-atomic 3D structure" is not accurate since atomic resolution is $\sim 1.0 \text{ \AA}$ (please see <https://journals.iucr.org/d/issues/2017/04/00/jm5025/>).

COMMSBIO-20-1826-T

Structure of the p53/RNA polymerase II assembly

Shu-Hao Liou¹, Sameer K. Singh¹, Robert H. Singer^{1,2}, Robert A. Coleman^{1*}, and Wei-Li Liu^{1*}

Response to reviewers comments

Reviewer #1

Reviewer #1 (Remarks to the Author):

In this manuscript, Wei-Li Liu and colleagues use single-particle cryo-EM in combination with computational modeling to deduce the structure of the human p53/Pol II complex to 4.6 Å resolution. The study is a follow-up of the authors' G&D 2016 paper (ref. 22) that determined 11 Å resolution cryo-EM structure of the same complex. The principal advances reported here consist of detection of conformational changes in the Pol II clamp coiled-coil, presence of a 17 Å wide cavity between the DNA-binding surface of p53 and Pol II, and localization of the TADs in between the Pol II jaw contributed by the Rpb1 and Rpb5 subunits. While these additional insights into p53/Pol II structure are not substantial, they do clarify how p53 might stimulate PIC formation. The paper has outstanding graphics; the molecular structures go well beyond those presented in their previous publication.

We appreciate this reviewer's insightful comments as well as the recognition of the potential significance regarding this report and we have revised our manuscript as suggested. In particular, we have now included several new statements and Figures as recommended as well as revisions to some text to clarify potential ambiguities. Please refer to the following points.

Principal concerns

1. If p53 binding induces movement of Pol II clamp into a further-closed state, it's unclear that the interaction between p53 and Pol II will enhance the efficiency of Pol II elongation as claimed. Is there any evidence of co-localization of p53 and Pol II within transcribed regions in mammalian cells? More thoughtful discussion on this point is required.

We thank the reviewer for highlighting this important concept. We speculate that the further-closed state may be playing a role in inhibiting non-specific loading of a p53/Pol II co-complex onto DNA. This further-closed state would be incompatible with the clamp's position in an elongating Pol II due to steric clashes with DNA in Pol II's cleft. We are currently working on a DNA bound form of a p53/Pol II co-complex that we hope will give more insight into p53's ability to help load Pol II

onto DNA and potentially stimulate elongation. We have included further statements to accentuate the existence of the further-closed state in the non-DNA bound p53/Pol II complex in the discussion section (page 13). In addition, we have clarified our point that p53 likely induces a clamp conformation similar to that found in the DSIF/Pol II assembly, since both factors bind the same region of Pol II and stimulate transcription elongation in biochemical assays (pages 13 & 14).

There is evidence from low resolution ChIP-seq and ChIP-qPCR suggesting that p53 may be co-traversing with elongating Pol II across coding regions in mammalian cells (Borsos et. al., Scientific Reports 2017). However this evidence does not definitively prove co-localization with Pol II on coding regions. To our knowledge, there has been only a single study examining p53 and Pol II co-localization genome-wide at high-resolution in mammalian cells (Chang et. al, Cell Reports 2014). However, these ChIP-exo studies of p53 and Pol II in human cells focused exclusively on Pol II occupancy at p53 REs, where co-localization was extremely robust. Further bioinformatic analysis of the data set to examine co-localization in coding regions is beyond the scope of this manuscript.

2. An unusual aspect of this and the precursor study is the implied mechanism of p53-Pol II association, namely, that it's the DNA-free p53 protein (possibly monomeric) that initially binds Pol II and that this complex subsequently associates with DNA-bound p53. This hypothetical mechanism, which to my knowledge has no precedent, needs to be made clearer. A schematic would be helpful.

We would like to thank the reviewer for suggesting a schematic. We have included a new supplemental figure (Figure S6) to highlight a hypothetical mechanism for loading of a monomeric p53/Pol II complex onto a DNA-bound p53 and included further text in the discussion (pages 16 & 17).

Figure S6

Minor points

1. Manuscript was written as if the reader had just finished reading the authors' G&D 2016 paper. This is unlikely to be the case. Results section requires more explanation on how the p53/Pol II co-complex was assembled, how the figures were generated, among other details.

We have now added additional details on the assembly of the p53/Pol II co-complex along with other relevant details in the Results section (page 6).

2. While the ms. was accurately and clearly written for the most part, there are a few places where it was not. For example: p. 3, line 4, "initiation" not elongation; p. 3, second paragraph, line 1, "multiple" not multiplex; p. 4, lines 1-3, entire sentence (awkward construction); p. 8, lines 1-2 (p53 core domain bound to DNA per se does not activate transcription); p. 8, line 12, "the possibility", not a possibility; p. 8, second paragraph, line 1, "...relative to Pol II/DNA" not related to Pol II/DNA; p. 18, second paragraph, line 1, "microscope" not microscopy.

We have made the recommended suggestions in the text as indicated.

3. Recommend changing "Core" to the more commonly used "core domain" throughout ms.

We have now changed the term "Core" to core domain throughout the manuscript.

4. Miscitation: Ref. 38.

We apologize for the miscitation and have placed this citation in the proper context (page 4).

5. Figure 5: R282 is not indicated.

We have now added labeling to show the location of R282 in Figure 5.

Figure 5

Liou_Figure 5

Figure 5

Reviewer #2

Reviewer #2 (Remarks to the Author):

Liou et al. present high resolution structural characterization of the p53-Pol II complex using single particle cryo-EM, revealing new insights into the mechanism of p53-dependent gene regulation. The manuscript is well written and they have done a good job selecting figures that illustrate important features of the structure models. All details leading to the cryo-EM reconstructions, models and fittings are very clearly described. There is simply nothing to be complained about the technical approaches chosen by the authors. I highly recommend the publication of this nice piece of work.

We would like to thank Reviewer 2 for their appreciation of our work.

Reviewer #3

Reviewer #3 (Remarks to the Author):

In this manuscript Liou et al., describe the cryo-EM structure of an assembly between RNA polymerase II and p53. The authors detail the site of p53 association on RNA Pol II and highlight a set of conformational changes within the Pol II clamp and identify a potential DNA binding cavity between the p53 core and Pol II. Overall the paper is well written, and the insights gleaned from the structure are of broad interest to the field. However, there are a number of inconsistencies that should be addressed and data that need to be visible before this manuscript could be considered appropriate for publication.

MAJOR CONCERNS

The cryo-EM density presented in the manuscript do not convincingly and unambiguously support the atomistic models presented in the Figures. While it is certainly clear that the presented cryo-EM map is markedly improved on previous structures, it is unclear from the figures presented if the cryo-EM density supports the proposed model unambiguously. The authors are strongly encouraged to carefully rework the figure panels to include clear representations of the EM density to provide evidence to the reader that the maps support the modeled interactions discussed. Specific concerns are broken down into two sections below, firstly some general comments about the overall quality of the reconstruction/model followed by specific concerns regarding the highlighted interactions relevant to the manuscript.

We would like to thank this reviewer for their comments to strengthen our manuscript. We have addressed these concerns via additional text, Tables, and supplementary figures displaying clear representations of EM densities in the suggested regions to support the modeled interactions discussed.

Responses to specific comments are listed below.

1. General Map Quality

The authors should provide a main text table bearing the data collection, reconstruction, and model statistics. It is difficult to assess the quality of the reconstruction and the fit of the model without reference to the model resolution based on an FSC threshold of 0.5, the Map CC (box), and the Map CC (mask). This table is standard in the field for presenting cryo-EM data and should definitely be included. Furthermore, the authors have no PDB code, suggesting that the provided map/model has not yet been curated by the PDB beyond the preliminary validation. This should be completed and the final validation report provided before further revision of the manuscript.

The Euler angle distribution suggests that the reconstruction suffers from preferred orientation, potentially explaining the streaky nature of the density in certain orientations. Could the authors comment on their confidence in the assigned interactions with Pol II given the limited orientations used for 3D reconstruction?

We would like to thank the reviewer for pointing out our discrepancy regarding the lack of a table with all indicated statistics. We have now included a table with this information in the results section (page 6). We have also included a PDB code (PDB: 6XRE) that has been curated and the final validation report as evidence of further validation in addition to an EMDB entry ID 22294.

Table 1

Cryo-EM data collection, refinement and validation statistics

	p53-Pol II (EMDB- 22294) (PDB 6XRE)
Data collection and processing	
Magnification	38168
Voltage (kV)	300
Electron exposure (e ⁻ /Å ²)	1.16
Defocus range (μm)	-0.5 - -2.5
Pixel size (Å)	0.655
Symmetry imposed	N/A
Initial particle images (no.)	776710
Final particle images (no.)	92522
Map resolution (Å)	4.6
FSC threshold	0.143
Map resolution range (Å)	3-11
Refinement	
Initial model used (PDB code)	5IYA/3TS8
Model resolution (Å)	4.6
FSC threshold	0.143

Model resolution range (Å)	3-11
Map sharpening B factor (Å ²)	6.216
Model composition	
Non-hydrogen atoms	34262
Protein residues	4289
Ligands	9
B factors (Å ²)	
Protein	12.66
Ligand	16.00
R.m.s. deviations	
Bond lengths (Å)	0.011
Bond angles (°)	1.309
Validation	
MolProbity score	1.75
Clashscore	4.76
Poor rotamers (%)	0.24
Ramachandran plot	
Favored (%)	91.53
Allowed (%)	7.89
Disallowed (%)	0.59
CC(box)/ CC(mask)	0.62/0.48

Table 1: Cryo-EM data collection, refinement and validations statistics

We also appreciate this reviewer's critique regarding a preferred orientation. As viewed in our Euler plot (Supplemental Figure S2), we have a sufficiently large number of particles encompassing views of p53 interacting with Pol II. This large number of particles has allowed us to achieve a selectively higher resolution of the p53 core domain relative to other regions of Pol II not interacting with p53. Thus, we are confident that the number of particles showing limited views of the p53/Pol II interaction is sufficient to achieve our expected resolution limits in this region.

2. Specific Interactions

2.1 A number of regions within p53 are re-modelled based on the cryo-EM map into conformations not previously observed within NMR, X-ray or Cryo-EM structures. Since the EM density is the main source of data in this study, the authors should include supplementary figures where the modified regions of interest are shown within the EM density (semi-transparent) so that the reader can evaluate the confidence of the modeling.

We have now included supplementary figures (Supplemental Figure S3B and S4A, pages 6-8) showing select regions of interest of our structure overlaid by semi-transparent EM density to support confidence in our modeling. We have included additional panels in these supplementary figures showing structural alignments of the core domain (Supplemental Figure S3A) and the individual helices of the TAD1/2 domain (Supplemental Figure S4B). These structural alignments indicate

that the majority of the p53 core domain in our p53/Pol II model remains structurally homologous to a DNA bound p53 (PDB: 3TS8). In addition, the individual helices within the TAD1/2 domain bound to Pol II are structurally homologous TAD1/2 helices bound to CBP from NMR studies (Supplemental Figure S4B).

Figure S3

A

B

Figure S4
A

B

C

2.2 One focus of the manuscript is the presence of a cavity between Pol II and the p53 core domain that is postulated to accommodate double-stranded DNA (Figure 2A). Based on published literature it appears reasonable that this site could accommodate the p53 response element DNA, especially when looking at panel 2B where there appears to be a substantial cleft that would reasonably accommodate dsDNA. However, looking at the cryo-EM density shown within Figure 2A there a number of spurious features within the map for which no protein is modelled that appear to occlude the proposed cavity. This is further reinforced in panel 4A where the cavity is not visible at all. Could the authors comment on the large amount of unmodelled density in this region? Are there regions within Pol II or p53 that could be explained by these densities?

There are two possibilities for regions within p53 and Pol II that could explain unmodeled densities in proximity to the cleft. First, the unmodeled density in this region could represent the extreme c-terminal region of p53 which extends from the oligomerization domain in the immediate vicinity (Supplemental Figure S3B, shown above in response to critique 2.1). This c-terminal portion of p53 is unstructured on its own but becomes structured upon binding interacting partners (Reviewed in Laptenko et. al., TIBS 2016). More importantly, this region of p53 is known to bind DNA non-specifically (Weinberg et. al., Journal of Molecular Biology 2004) placing it in very close proximity to the cavity that could accommodate dsDNA.

The CTD of Pol II's RPB1 subunit, which is highly unstructured, is also known to bind the core domain of p53 and therefore could be responsible for the extra unmodeled densities (Kim et. al., PLoS One, 2011).

2.3 Consistent with the above comment, the authors present a remodeled conformation of the Pol II clamp coiled coil which has been modelled 'further downwards'. However, looking at EM density presented in Figure 4A, it does not appear unambiguously convincing that this is appropriate. There are a lot of spurious features in the density that could be attributed to the closed loop conformation of the Pol II clamp. It is also possible that the 3D reconstruction suffers from heterogeneity caused by the Pol II clamp adopting a closed and open conformation. As a result, the map may represent an average of the Pol II clamp in an open and closed state which results in this smearing of the density. Have the authors considered 3D classification in RELION using partial particle subtraction to resolve this? This could both improve the quality of the density and serve to provide higher resolution of the interfaces.

We would like to thank the reviewer for suggesting 3D classification in RELION as there does appear to be some flexibility issues in this region. To better resolve both Pol II and p53 density, a Relion-implemented multibody refinement involving particle subtraction was performed. To prepare both Pol II and p53 masks, a 6 Å resolution map of human closed-form Pol II (RCSB number 5IYA) was used to subtract p53/Pol II density to generate a p53 volume. Pol II and p53 masks were

generated by `relion_mask_create`, with an extended 5 pixels of binarized maps, 5 pixels of soft edge and 10 Å of lowpass filter. A refinement with 0.9 degree of sampling was performed and p53 was assigned to rotate corresponding with Pol II. The standard deviations of angular and translational prior for Pol II and p53 are 5 and 6 pixels, and 1 and 3 pixels, respectively. This multibody refinement strategy significantly improves the EM density of the Pol II clamp (Supplemental Figure S5). More importantly, there is no significant density in the immediate region where an open or closed clamp would be predicted to exist. Rather the density of the clamp has shifted downward where it did not previously exist in structures of open and closed Pol II. Therefore our interpretation of the data is that the clamp has adopted a new further-closed conformation.

Figure S5

2.4 Finally, the N-terminal transactivation domain of p53, TAD1/2, are modelled interacting with the Rpb1, Rpb2, and Rpb5 subunits of RNA Pol II (Figure 3A). This is an instance when the authors should present a detailed cryo-EM map showing the modelled regions within the density so that the reader can evaluate the proposed interaction from the actual data.

We thank the reviewer for pointing out the importance of the interaction between Pol II and the TAD1/2 domain. We are confident about TAD's position between the jaws of Pol II based on the following evidence. This additional density between Pol II's jaws was not observed in Pol II's cryo-EM structure from additional groups (Bernecky et. al., Nature 2016 and He et. al, Nature 2016). The additional density is also approximately the same size as expected for p53's TADs (Supplemental Figure S4A, shown above in response to critique 2.1). In addition, our previous biochemical results from label transfer assays provides strong support that p53's N-terminus is in close proximity to the Rpb1 subunit of Pol II (Singh et al., Genes & Dev 2016).

Analysis using Resmap reveals that the resolution of this extra density is ~5-7 Angstroms. Based on these above facts, we used Rosetta to model the TAD1/2 domain into this density (Supplemental Figure S4A, shown above in response to critique 2.1). Although this region displays mediocre resolution, we feel comfortable assigning TAD1(17-43aa) and TAD2 to two sausage-like density portions. Even when we increase the threshold to 0.0112, the structures are still well aligned within the density. Albeit, TAD2 may be slightly outside of the density for stability reasons as suggested by Rosetta.

Our model is consistent with multiple NMR structures of TAD1/2 (PDB: 2L14 and 5HOU) bound to different regions of CBP showing overall conservation of the individual TAD1/2 helices comprising this domain (Supplemental Figure S4B, shown above in response to critique 2.1). However, the orientations between the TAD1/2 helices within this domain are structurally heterogenous amongst all structures (Supplemental Figure S4C, shown above in response to critique 2.1) likely indicating an interaction specific co-folding of the TAD1/2 domain. As a result, we acknowledge that the loops connecting the helices within this domain are not well defined. We have revised the results (page 10) section accordingly to highlight these findings.

Minor Comments

Page 12 – “near-atomic 3D structure” is not accurate since atomic resolution is ~1.0 Å (please see <https://journals.iucr.org/d/issues/2017/04/00/jm5025/>).

We appreciate the reviewer's comments and have changed the wording of near-atomic 3D structure in the manuscript accordingly (page 12).

REVIEWERS' COMMENTS:

Reviewer #1 (Remarks to the Author):

My concerns have been satisfactorily met.

A few minor grammatical issues remain:

p. 2, last line: "insights into p53-regulated gene expression."

p. 3, line 4. "PIC formation", not "The PIC formation."

p. 12, line 1: "Recent advances in single particle..."

p. 13, last line: "Pol II clamp conformation..." (not confirmation)

p. 18, line 1: "PIC assembly", not "the PIC assembly".

COMMSBIO-20-1826-T
Structure of the p53/RNA polymerase II assembly

Shu-Hao Liou¹, Sameer K. Singh¹, Robert H. Singer^{1,2}, Robert A. Coleman^{1*}, and Wei-Li Liu^{1*}

Response to reviewers comments

Reviewer #1

Reviewer #1 (Remarks to the Author):

Q: A few minor grammatical issues remain:

- p. 2, last line: "insights into p53-regulated gene expression."
- p. 3, line 4. "PIC formation", not "The PIC formation."
- p. 12, line 1: "Recent advances in single particle..."
- p. 13, last line: "Pol II clamp conformation..." (not confirmation)
- p. 18, line 1: "PIC assembly", not "the PIC assembly".

A: We have now made changes in the text based on Reviewer #1's comments.